# Relationship between COVID-19 Mortality, Hospital Beds, and Primary Care by Italian Regions: A Lesson for the Future

**DOI:** 10.3390/jcm11144196

**Published:** 2022-07-19

**Authors:** Nicola Ferrara, Carlo Pietro Campobasso, Sergio Cocozza, Valeria Conti, Sergio Davinelli, Maria Costantino, Alessandro Cannavo, Giuseppe Rengo, Amelia Filippelli, Graziamaria Corbi

**Affiliations:** 1Department of Translational Medical Sciences, University of Naples Federico II, 80131 Naples, Italy; nicola.ferrara@unina.it (N.F.); alessandro.cannavo@unina.it (A.C.); giuseppe.rengo@unina.it (G.R.); 2Istituti Clinici Scientifici ICS-Maugeri, 82037 Telese Terme, Italy; 3Department of Experimental Medicine, University of Campania Luigi Vanvitelli, 80138 Naples, Italy; carlopietro.campobasso@unicampania.it; 4Department of Molecular Medicine and Medical Biotechnology, University of Naples Federico II, 80131 Naples, Italy; sergio.cocozza@gmail.com; 5Department of Medicine, Surgery and Dentistry “Scuola Medica Salernitana”, University of Salerno, 84081 Baronissi, Italy; vconti@unisa.it (V.C.); mcostantino@unisa.it (M.C.); afilippelli@unisa.it (A.F.); 6Department of Medicine and Health Sciences, University of Molise, 86100 Campobasso, Italy; sergio.davinelli@unimol.it

**Keywords:** COVID-19, SARS-CoV-2 infection, mortality, healthcare system, community healthcare, hospital healthcare

## Abstract

One of the characteristics of the SARS-CoV-2 infection in Italy is the significant regional difference in terms of lethality and mortality. These geographical variances were clear in the first wave and confirmed in the second one as well. The study aimed to analyze the correlation between regional differences in COVID-19 mortality and different regional care models, by retrospectively analyzing the association between the Italian COVID-19 deaths and the number of hospital beds, long-term care facilities, general practitioners (GPs), and the health expenditure per capita. The period considered was from 1 March 2020 to 1 March 2021. The number of hospital beds (*p* < 0.0001) and the number of GPs (*p* = 0.0094) significantly predicted the COVID-19 death rate. The Italian regions with a higher number of hospital beds and a lower number of GPs showed a higher number of deaths. Multivariate analyses confirmed the results. The Italian regions with a higher amount of centralized healthcare, as represented by the number of hospital beds, experienced a higher number of deaths, while the regions with greater community support, as exemplified by the number of the GPs, faced higher survival. These results suggest the need for a change in the current healthcare system organization.

## 1. Introduction

In February 2021, the number of confirmed cases of the COVID-19 pandemic worldwide was more than one hundred million, of which more than two million had led to the death of the patient. In the same period, more than two million positive cases of SARS-CoV-2 infection were recorded in Italy, making our country the eighth in the world and the fourth in Europe for total cases [1]. Italy was also the sixth deadliest country in the world and the second in Europe for the number of deaths with more than ninety-one thousand victims.

One of the characteristics of the SARS-CoV-2 infection observed in Italy is the large regional difference in terms of lethality and mortality due to COVID-19 [1]. These geographical differences were very clear in the first wave and partially confirmed in the second one as well [2]. The first death due to severe acute respiratory syndrome infection in a COVID-19 positive patient was observed in the Lombardy region (the highest populated among the 21 Italian regions) on 21 February 2020. Since February, the municipalities located in Northern Italy were the Italian regions most severely hit by the pandemic up to 31 May 2020, as represented by a noticeable or substantial excess in overall mortality [1]. A new rise in the pandemic curve was observed from September 2020 after the end of the summer holidays. Significant outbreaks began to increase in number, starting in the largest Italian cities such as Milan, Rome, and Naples in Southern Italy [3].

Italy’s first response to COVID-19 has been described in detail as an unprecedented challenge by the World Health Organization (WHO) in a 2020 report [4]. In the WHO report, it is emphasized that the Italian healthcare system is a regionally based national health system, and highly decentralized. The Italian Constitution makes health protection a shared responsibility between the central government and the 21 Italian regions. The central government legislates on the principles assessed in the Italian Constitution, but the regions can adapt them to their local contexts and have the budgets to manage healthcare delivery through local health agencies [4]. In fact, the National Health System in Italy has been strongly regionalized for over 30 years with substantial regional differences in terms of management, financing, objectives, and models adopted.

Despite such a decentralized healthcare system and the stagnation of the Italian economy and growth for decades compared to other European and most developed countries, the world’s healthiest people were those living in Italy, after Spain, based on the Bloomberg 2019 Healthiest Country Index [5]. The National Healthcare Service has faced increasing pressure during the COVID-19 pandemic.

The aim of this study is to analyze whether any correlation between regional differences in COVID-19 mortality and different regional care models, levels of funding, and socio-economic and demographic features occurred.

## 2. Materials and Methods

A retrospective study was conducted. The data source was represented (i.e., regional data for the resident population, the aging index, the birth life expectancy, the number of hospital beds, the number of long-term care facilities (LTCF), the number of the general practitioners (GPs), the health expenditure per capita, the percentage of over 65 years old subjects) by the database of the Italian Institute of Statistics (ISTAT), with the last available records [6]. The COVID-19 death rate was derived by the Italian Health Ministry and Istituto Superiore di Sanità (ISS) database [7].

The period considered was from 1 March 2020 to 1 March 2021.

The study was performed following the Guidelines for Accurate and Transparent Health Estimates (GATHER) 2016 (see supplemental material).

### Statistical Analysis

The distribution of data was analyzed by using the Skewness and Kurtosis normality tests. All the considered variables showed a normal distribution. Then, to evaluate the association between the COVID-19 mortality rate and the considered parameters, a univariate analysis was firstly performed, by matching the variables one-by-one. Then, multiple linear regression analyses were performed to identify the possible predictors of the COVID-19 death rate. The mortality rate was used as a dependent variable, while all the other considered parameters were used as independent factors. Because of the risk of collinearity, the aging index (that is the number of elderly populations ≥65 years old per 100 persons younger than 14 years old), the life expectancy, and the percentage of over 65 years old subjects were introduced one by one in the analysis. Therefore, 3 different models of the multivariate linear regression analysis were constructed. A *p*-value < 0.05 was considered statistically significant. All the analyses were performed by using STATA 16 statistical software (StataCorp. 2019. Stata Statistical Software: Release 16. College Station, TX, USA: StataCorp LLC).

## 3. Results

Table 1 reports the distribution of COVID-19 total deaths and COVID-19 deaths adjusted for the number of inhabitants in 21 Italian regions.

Primarily, a simple linear regression was used to test if each variable significantly predicted the COVID-19 death rate (per 10,000). Our analysis revealed two main findings.

Firstly, it was found that the number of hospital beds (per 1000) significantly predicted the COVID-19 death rate (β = 17.64346, 95% CI 9.740126 25.5468, *p* < 0.0001). The Italian regions with a higher number of hospital beds (×1000 inhabitants) showed a higher number of deaths. The fitted regression model was: COVID-19 death rate = −40.49563 + 17.64346 * (N of hospital beds per 1000). The overall regression was statistically significant (r2 = 0.5347, F (1, 19) = 21.83, *p* = 0.0002).

Secondly, the number of general practitioners (GPs) (per 10,000 inhabitants) significantly predicted the COVID-19 death rate (β = −33.48067, 95% CI −57.7291 −9.232238, *p* = 0.0094): the Italian regions with a higher number of GPs showed a lower number of COVID-19 deaths. The fitted regression model was: COVID-19 death rate = 45.84148–33.48067 * (N of GPs per 10,000 inhabitants). The overall regression was statistically significant (r2 = 0.3053, F (1, 19) = 8.35, *p* = 0.0094). No associations were found between COVID-19 mortality rate and health expenditure per capita, life expectancy, aging index, and percentage of over 65 years old subjects per Italian region in the univariate analyses.

To assess the best predictors of COVID-19 mortality rate among the considered parameters, three different multivariate linear regression analyses were performed to overcome the risk of collinearity with the percentage of 65 years old subjects, the aging index, and the life expectancy. The COVID-19 mortality rate was used as a dependent variable in all models. The results of the three models are shown in Table 2.

All three models confirmed the results of the univariate analyses. In all three models, the number of hospital beds (×1000 inhabitants) represented the best predictors of the COVID-19 mortality rate, with a direct relationship between the number of deaths and the number of hospital beds, and an inverse relationship between the number of COVID-19 deaths and number of GPs was found. These findings suggest that the Italian regions with a higher amount of centralized healthcare, as represented by the number of hospital beds, experienced a higher number of deaths, while the regions with greater community support, as represented by the number of the GPs, faced higher survival (Table 2).

In the first model, the multiple linear regression analysis tested if the percentage of subjects ≥65 years old, the number of hospital beds (per 1000), the health expenditure per capita, the GPs per 1000, and the number of LTCF significantly predicted the COVID-19 death rate (per 100,000). The fitted regression model was: −13.01125 + 0.6861688 * (% of over65 yrs old) + 14.83807 * (N of hospital beds per 1000) − 14.83807 * (Health expenditure per capita) − 33.64296 * (N of GPs per 1000) + 0.0034141 * (N of LTCF).

The overall regression was statistically significant (r2 = 0.8373, F (5,13) = 13.38, *p* = 0.0001). It was found that the number of hospital beds (×1000 inhabitants) significantly predicted the COVID-19 death rate (per 100,000) (β = 14.83807, 95%CI 5.634418 24.04171, *p* = 0.004) (Table 2).

The Italian regions with a higher number of hospital beds (×1000 inhabitants) showed a higher COVID-19 death rate (Figure 1A). Similarly, the number of GPs also predicted the COVID-19 death rate (per 100,000) (β = −33.64296, 95%CI −53.6297 −13.6563, *p* = 0.003). The Italian regions with a higher number of GPs showed a lower COVID-19 death rate (Figure 1B).

Then, the second multiple linear regression tested if the life expectancy, the number of hospital beds (per 1000), the health expenditure per capita, the GPs per 1000, and the number of LTCF significantly predicted the number of COVID-19 deaths (per 100,000 inhabitants). The fitted regression model was: 94.9212–1.3861 * (life expectancy) + 15.4037 * (N of hospital beds per 1000) + 0.0090 * (health expenditure per capita) − 32.4074 * (N of GPs per 1000) + 0.0049 * (N of LTCF).

The overall regression was statistically significant (r2 = 0.8292, F (5,13) = 12.62, *p* < 0.0001). It was found that again the number of hospital beds (×1000 inhabitants) (β = 14.83807, 95%CI 26.0161 24.7913, *p* = 0.004), and the number of GPs (β = −32.4074, 95%CI −53.31456 −11.5003, *p* = 0.005), significantly predicted the number of COVID-19 deaths (per 100,000 inhabitants) (Table 2).

The Italian regions with a higher number of hospital beds (×1000 inhabitants, Figure 2A) and a lower number of GPs (per 1000), Figure 2B) experienced a higher number of COVID-19 deaths.

Finally, the third multiple linear regression tested if the aging index, the number of hospital beds (per 1000), the health expenditure per capita, the GPs per 1000, and the number of LTCF significantly predicted the number of COVID-19 deaths (per 100,000 inhabitants). The fitted regression model was: −4.7326 + 0.0413 * (aging index) + 15.5586 * (N of hospital beds per 1000) − 0.0022 (health expenditure per capita) − 34.8603 (N of GPs per 1000) + 0.0032 (N of LTCF).

The overall regression was statistically significant (r2 = 0.8292, F (5,13) = 12.62, *p* = 0.0001). It was found that the number of hospital beds (×1000 inhabitants) significantly predicted the number of COVID-19 deaths (per 100,000) (β = 15.5587, 95%CI 6.1525 24.9648, *p* = 0.003) (Table 2).

The Italian regions with a higher number of hospital beds (×1000 inhabitants) showed a higher COVID-19 death rate (Figure 3A). Similarly, the number of GPs also predicted the COVID-19 death rate (per 100,000) (β = −34.86028, 95%CI −55.43874 −14.28183, *p* = 0.003). The Italian regions with a higher number of GPs showed a lower COVID-19 death rate (Figure 3B).

## 4. Discussion

In March 2020, the COVID-19 outbreak spread across all of Italy. From 1 March 2020 to 1 March 2021, about 3 million positive confirmed cases and over 100,000 deaths were recorded in Italy [8]. Large differences in COVID-19 mortality have been demonstrated between the 21 Italian regions. Saglietto and collaborators [9], analyzing the number of deaths related to COVID-19 in relation to the number of tests performed for SARS-CoV-2 among the most affected Italian regions, showed that the Italian regions with the highest number of tests had the lowest rate of mortality per 100,000 population. This evidence suggested that early identification and isolation of active cases (including asymptomatic or mildly symptomatic subjects) could have had an important effect in reducing COVID-19 mortality, by hypothesizing that these differences could explain the wide differences in mortality among the Italian regions [9]. Moreover, Saglietto et al. [9] also underlined that different COVID-19 mortality experienced by Italian regions was only partially explained by differences in the population age structure.

Some additional confounding factors, not analyzed in our study, could limit the real impact of these results on the different COVID-19 mortality among the Italian regions. During the first and the second wave of the pandemic, the number of autopsies dramatically declined in Italy due to the concerns about the management of corpses who died by or with COVID-19, as also included in the first recommendations provided by the Italian Health Ministry to pathologists and morgue workers [10]. The excess of mortality could be higher (maybe triple) than the one reported in the official epidemiological surveys [11,12]. Missing autopsies as well as false negative cases of antigenic and molecular tests of first generation may have produced a distorting effect on the assessment of the real mortality rate and the excess mortality [13].

Analyzing the data recorded by the Italian Health Ministry, we found a strong direct association between COVID-19 deaths and the number of hospital beds (*p* = 0.003): the Italian regions with a higher number of hospital beds (x 1000 inhabitants) showed a higher number of deaths, and this evidence was confirmed by all the multiple regression analyses we performed (Figure 1, Figure 2 and Figure 3 panels A, Table 2).

This finding could be related to the greater severity of the COVID-19 patients, which was not considered in this study because of the lack of information provided from the official report of the Italian Health Ministry, concerning the clinical history of the victims by regional distribution.

However, the absence of a reliable statistical relationship between the aging index, or percentage of over 65 years old subjects (Appendix A), the most affected by comorbidity, and the deaths support the hypothesis that the different regional mortality rates should be also ascribed to different management choices for the organization and delivery of health services. The coronavirus pandemic exposed the weaknesses and dysfunctionality of our current decentralized healthcare system. According to Armocida et al. [14], healthcare systems’ capacity and financing need to be more flexible to take into account exceptional emergencies, and stronger national coordination should be in place to provide more sustainable and efficient health services.

Then, our findings suggest that the choice to concentrate the healthcare services in the hospitals should be reconsidered toward a more community-centered approach, as was recently also recommended by the World Health Organization [15]. In most populated European countries, even if the hospital beds were efficiently managed, the concentration of hospitalized patients and the scarcity of beds have put pressure on the hospital systems [16]. Considering the use of the hospital resources, Pecoraro et al. [16] demonstrated that, whereas Germany counts a high availability of hospital beds per 100,000 inhabitants, it is the country with the lowest number of hospitalizations in the general wards (N = 40.8) compared with the other countries (Italy, N = 91.1; France, N = 105.4; and Spain, N = 220.7). The authors underlined how this finding was associated with lower mortality in Germany (where the number of deaths per 100.000 inhabitants was 10.9 in contrast to the data reported in France (N = 30.0), Italy (N = 57.8), and Spain (N = 60.4) [16].

In our study, the need for a more community-centered approach seems to be also supported by the strong association (confirmed by the multivariate linear regression analyses) between COVID-19 deaths (×10,000 inhabitants) and the number of GPs (per 10,000 inhabitants) per Italian region. In all models: the Italian regions with a higher number of GPs showed a lower number of COVID-19 deaths (Figure 1, Figure 2 and Figure 3 panels B, Table 2).

Recently, the great interest in this topic was also shown by the discussion on the national and regional government decisions taken to counteract COVID-19 dissemination [13]. Italy has mistreated its primary healthcare system in the last decades with an availability of 89 GPs per 100,000 inhabitants, compared to 167. in Germany, and 155.5 in France [17,18].

According to Plagg et al. [18], increasing the number of available hospital beds without appropriate staff and reinforcement of the primary care services cannot be considered the answer, as represented by increased mortality.

As reported in a letter in the Lancet Journal [19], the contribution to documenting and analyzing the social and political effects of epidemiological events has been crucial for other infectious diseases. Sams et al. [19] underlined that, as in the occasion of other epidemics (e.g., Ebola virus disease and AIDS), networks have become central to addressing issues such as vaccine hesitancy, misinformation, and trust; all problems also experienced in the Italian pandemic [20].

### Limitations and Strengths

The present study suffers some limitations. Firstly, because of the retrospective design, the analyzed data did not permit the definition of the COVID-19 severity, and the impact of other factors on the total mortality rate. In particular, the impact of hospitalization on mortality, and the viral characteristics, such as the SARS-CoV-2 viral load and replication rate, could represent other possible confounders.

However, the used data represent the official ones provided by the Italian Health Ministry, and, to our knowledge, this represents the first study that analyzes the relationship between the number of GPs and COVID-19 mortality rate, accordingly also to the hospital bed number.

## 5. Conclusions

This study analyzes, for the first time, the relationship between COVID-19 mortality rates, the number of hospital beds, the number of GPs, and, therefore, the regional healthcare system organization. In the future, a more flexible hospital system cooperating with primary medical care might show better efficiency in containing viral spread and managing patients. Noteworthily, the hospitals still represent the most reliable resource of healthcare assistance in Italy, and our findings do not advise against hospitalization but encourage better management and optimization of healthcare sources.

Moreover, a better understanding of the social and political factors affecting the outcomes of pandemic emergencies should represent a priority, also for the increasing number of elderly, and a rise in healthcare requests. Surely further studies are needed to better clarify these aspects, also considering the different distribution of healthcare sources by country.

## Figures and Tables

**Figure 1 jcm-11-04196-f001:**
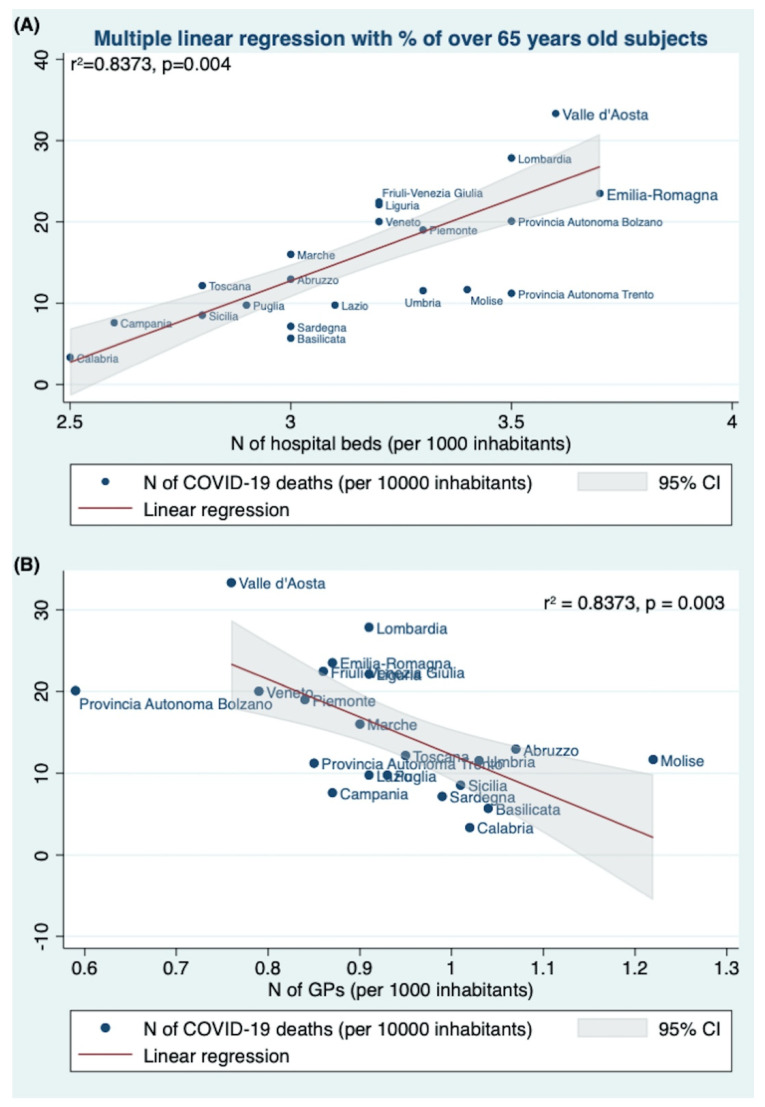
Multiple linear regression among the number of COVID-19 deaths (×10,000 inhabitants) and (**A**) the number of hospital beds (×1000 inhabitants), and (**B**) the number of GPs (×1000 inhabitants) per Italian region. In a multivariate linear regression analysis, introducing the number of COVID-19 deaths (per 100,000) as the dependent variable and the percentage of subjects ≥65 years old, the number of hospital beds (per 1000), the health expenditure per capita, the GPs per 1000, and the number of LTCF significantly as the independent variable, the number of hospital beds (per 1000) (β = 14.8381, *p* = 0.004) and the number of GPs (per 1000 inhabitants) (β = −33.6430, *p* = 0.003) significantly predicted the number of COVID-19 deaths (×10,000 inhabitants). The Italian regions with a higher number of hospital beds (×1000 inhabitants) and a lower number of GPs showed a higher COVID-19 death rate. The line shows the linear regression value obtained by the multivariate analysis.

**Figure 2 jcm-11-04196-f002:**
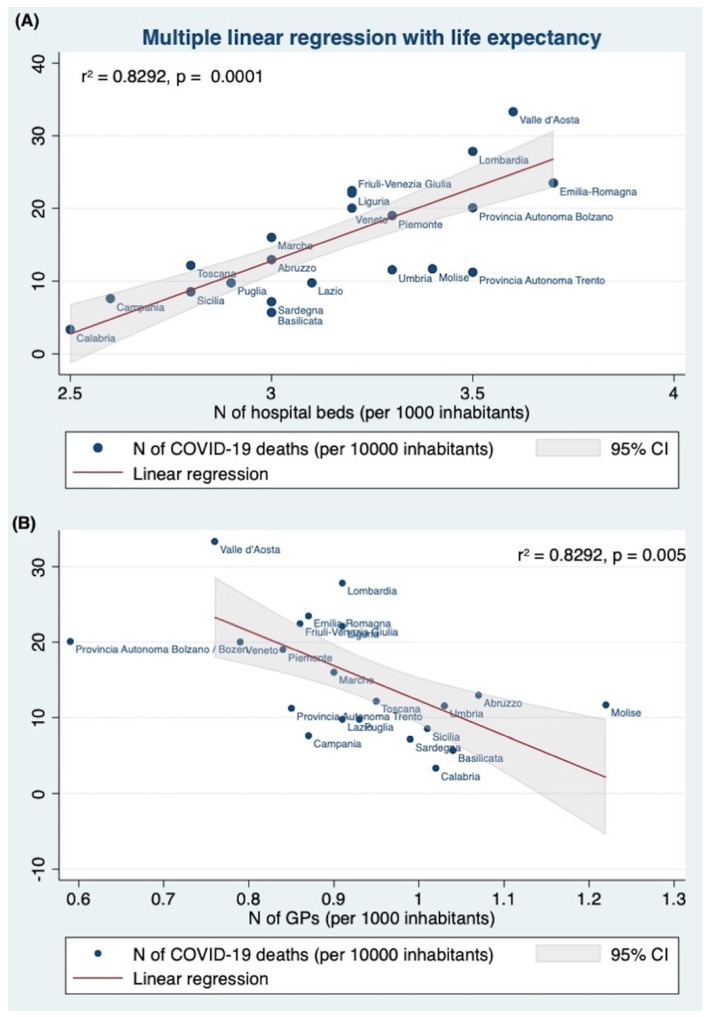
Multiple linear regression among the number of COVID-19 deaths (×10,000 inhabitants) and (**A**) the number of hospital beds (×1000 inhabitants), and (**B**) the number of GPs (×1000 inhabitants) per Italian region. In a multivariate linear regression analysis, introducing the number of COVID-19 deaths (per 100,000) as the dependent variable and the life expectancy, the number of hospital beds (per 1000), the health expenditure per capita, the GPs per 1000, and the number of LTCF significantly as the independent variable, the number of hospital beds (per 1000) (β = 15.4037, *p* = 0.004) and the number of GPs (per 1000 inhabitants) (β = −32.4074, *p* = 0.005) significantly predicted the number of COVID-19 deaths (×10,000 inhabitants). The Italian regions with a higher number of hospital beds (×1000 inhabitants) and a lower number of GPs showed a higher COVID-19 death rate. The line shows the linear regression value obtained by the multivariate analysis.

**Figure 3 jcm-11-04196-f003:**
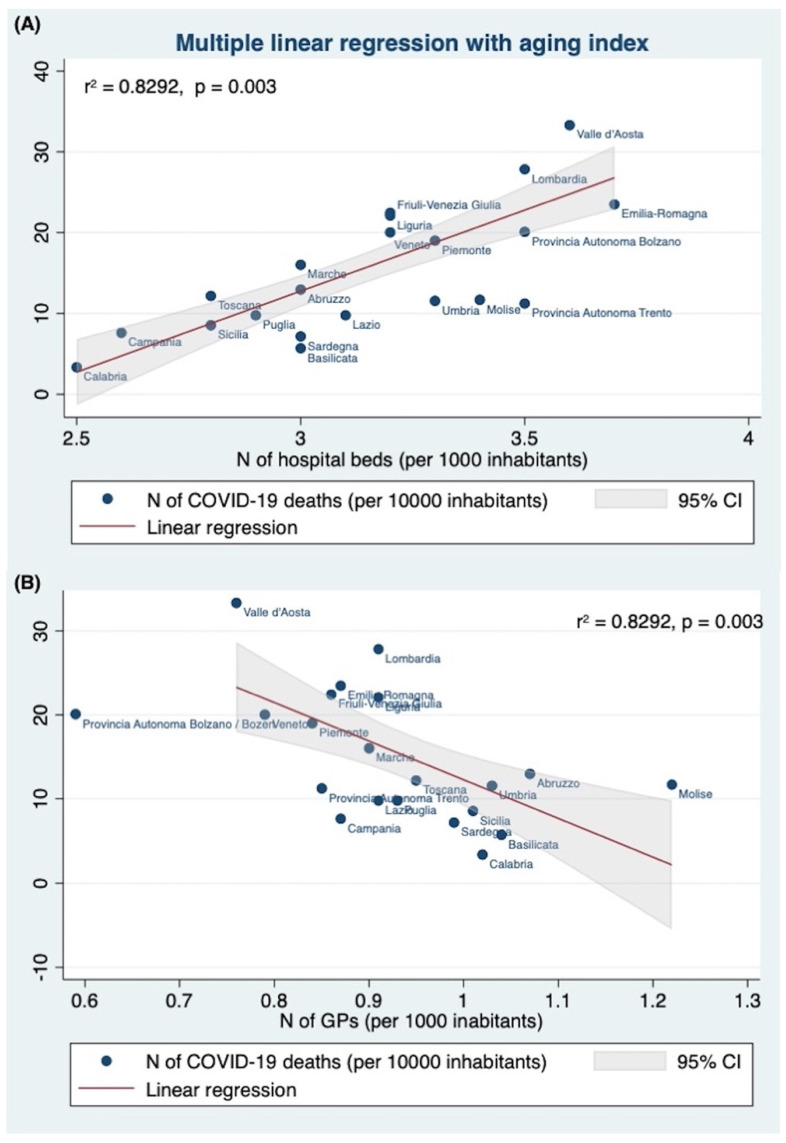
Multiple linear regression among the number of COVID-19 deaths (×10,000 inhabitants) and (**A**) the number of hospital beds (×1000 inhabitants), and (**B**) the number of GPs (×1000 inhabitants) per Italian region. In a multivariate linear regression analysis, introducing the number of COVID-19 deaths (per 100,000) as the dependent variable and the aging index, the number of hospital beds (per 1000), the health expenditure per capita, the GPs per 1000, and the number of LTCF significantly as the independent variable, the number of hospital beds (per 1000) (β = 15.55866, *p* = 0.003) and the number of GPs (per 1000 inhabitants) (β = −34.8603, *p* = 0.003) significantly predicted the number of COVID-19 deaths (×10,000 inhabitants). The Italian regions with a higher number of hospital beds (×1000 inhabitants) and a lower number of GPs showed a higher COVID-19 death rate. The line shows the linear regression value obtained by the multivariate analysis.

**Table 1 jcm-11-04196-t001:** The distribution of COVID-19 total deaths, and COVID-19 deaths adjusted for the number of inhabitants in 21 Italian regions.

Italian Regions	Regional Inhabitants (×1000)	COVID-19 Total Deaths	Deaths (×10,000 Inhabitants)
**Valle d’Aosta**	125.7	419	33.333
**Lombardia**	10,060.6	28,006	27.8373
**Emilia Romagna**	4459.5	10,472	23·4824
**Friuli-Venezia Giulia**	1215.2	2726	22.4325
**Liguria**	1550.6	3426	22.0947
**Bolzano (Provincial area)**	531.2	1067	20.0866
**Veneto**	4905.9	9822	20.0208
**Piemonte**	4356.406	8276	18.9973
**Marche**	1525.3	2442	16.0099
**Abruzzo**	1311.6	1699	12.9536
**Toscana**	3729.6	4539	12.1702
**Molise**	305.6	357	11.6819
**Umbria**	882	1019	11.5533
**Trento (Provincial area)**	1072.3	1204	11.2282
**Puglia**	4029.1	3938	9.7740
**Lazio**	5879.1	5745	9.7719
**Sicilia**	4999.9	4272	8.5442
**Campania**	5801.7	4420	7.6185
**Sardegna**	1639.6	1176	7.1725
**Basilicata**	562.9	321	5.7026
**Calabria**	1947.1	654	3.3588

**Table 2 jcm-11-04196-t002:** Models of multivariate linear regression analyses.

	Model 1				Model 2				Model 3			
COVID-19 Deaths (×10,000 Inhabitants)	Beta	95% CI		*p*	Beta	95% CI		*p*	Beta	95% CI		*p*
		Low	High			Low	High			Low	High	
Subjects ≥ 65 yo (%)	0.69	−0.59	1.96	0.266	---	---	---	---	---	---	---	---
Hospital beds (per 1000)	14.84	5.63	24.04	0.004	15.40	6.02	24.79	0.004	15.56	6.15	24.96	0.003
Health expend. pc	−0.001	−0.03	0.03	0.902	0.01	−0.02	0.03	0.504	−0.01	−0.04	0.03	0.901
GPs per 1000	−33.64	−53.63	−13.66	0.003	−32.41	−53.31	−11.50	0.005	−34.86	−55.44	−14.28	0.003
Number of LTCF	0.003	−0.01	0.01	0.520	0.01	−0.01	0.02	0.422	0.01	−0.01	0.01	0.557
Life expectancy	---	---	---	---	−1.39	−5.04	2.27	0.427	---	---	---	---
Aging index	---	---	---	---	---	---	---	---	0.04	−0.07	0.15	0.427

CI, confidence interval; yo, years old; Health expend. Pc, health expenditure per capita; GPs, general practitioners; LTCF, long-term care facilities. Model 1 included as covariates the percentage of subjects ≥ 65 years old, number of hospital beds, health expenditure, number of GPs, and number of LTCF. Model 2 included, as covariates, the number of hospital beds, health expenditure, number of GPs, number of LTCF, and life expectancy. Model 3 included, as covariates, the number of hospital beds, health expenditure, number of GPs, number of LTCF, and the aging index.

## Data Availability

The data are available on request to the corresponding author.

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
