# Peer review of "Relationship between COVID-19 Mortality, Hospital Beds, and Primary Care by Italian Regions: A Lesson for the Future"

_jcm, 2022, doi:10.3390/jcm11144196_

Round 1
Reviewer 1 Report
Thank you for the opportunity of reviewing the manuscript about the regional variance of COVID-19 mortality in Italy. I reviewed this manuscript with great interest. But there are some problems.
First, Model 1, 2 and 3 are considered in the analysis. The authors described “The aim of this study is to analyze whether any correlation between regional differences in COVID-19 mortality and different regional care models, levels of funding, and socio-economic and demographic features occurred.”
If some factors are used to predict COVID-19 mortality, it is important to consider various models to derive the best model. But if a multivariable linear regression model was performed to evaluate the relationship, the authors should put all confounding factors into one regression model. This is because all confounding factors exist simultaneously in reality.
Second, since patient’s age has a large effect on COVID-19 mortality, the authors should perform a subgroup analysis by age group. If it is difficult to evaluate the relationship by age group using existing data, then at least the elderly and the others should be evaluated separately.
Reviewer 2 Report
The authors analyzed the correlation between 22 regional differences in COVID 19 mortality and different regional care models, retrospectively ana-23 lyzing the association between the Italian COVID-19 deaths and the number of hospital beds, Long-24 Term Care Facilities, General Practitioners (GPs), and the health expenditure per capita. Their main findings are that Italian regions with a higher number of hospital beds (x 1000 inhabitants) showed a higher COVID-19 death rate, while the Italian regions with a higher number of GPs showed a lower COVID-19 212 death rate.
I appreciated the draft, found it rigorous and of interest and with a relatively novel core message. It could have an impact on strategical allocation of healthcare resources.
I only have some issues to stress:
1 The source of the data for the authors is given by ISTAT. However, for instance, another Italian group - Oliva et al. underlined that, especially in the first phase of the pandemic, the number of autopsies relatively declines and so there could have been missed diagnoses and misdiagnoses of COVID-19 biasing national data (doi: 10.1097/PTS.0000000000000793). This is a topic of outmost importance and the authors should comment it and explain whether and how much this fact could influence their observations.
2 In the statistical analysis section the kind of distribution should be made explicit and thus it should be clearly explained why a specific statistical approach was chosen.
3 Some considerations regarding the impact of hospitalization on the mortality (and on viral characteristics like SARS-CoV-2 viral load and replication) could be of significant interest, in particular to stress the fact that the data of this study do not advise against hospitalization.
4 I would add more comments on the comparisons with other countries, in order to stress the translational value of the found evidence.
Minor comments:
Lines 38-41 can be removed. Lines 49-51 deserve a clear discussion and reference(s). Line 99 name of the producer and country where the software is produced are missing. There are some choices of words that could be improved (eg. line 58: "largest" rather than "bigger").
Round 2
Reviewer 1 Report
I agreed the responses by the authors. There is no comments from this reviewer.